# Task-Focused Consolidation with Spaced Recall: Making Neural Networks Learn like College Students

## Abstract

Deep neural networks often suffer from a critical limitation known as catastrophic forgetting, where performance on past tasks degrades after learning new ones. This paper introduces a novel continual learning approach inspired by human learning strategies like Active Recall, Deliberate Practice, and Spaced Repetition, named Task-Focused Consolidation with Spaced Recall (TFC-SR). TFC-SR enhances the standard experience replay framework with a mechanism we term the Active Recall Probe. It is a periodic, task-aware evaluation of the model's memory that stabilizes the representations of past knowledge. We test TFC-SR on the Split MNIST and the Split CIFAR-100 benchmarks against leading regularization-based and replay-based baselines. Our results show that TFC-SR performs significantly better than these methods. For instance, on the Split CIFAR-100, it achieves a final accuracy of 13.17% compared to Standard Experience Replay's 7.40%. We demonstrate that this advantage comes from the stabilizing effect of the probe itself, and not from the difference in replay volume. Additionally, we analyze the trade-off between memory size and performance and show that while TFC-SR performs better in memory-constrained environments, higher replay volume is still more effective when available memory is abundant. We conclude that TFC-SR is a robust and efficient approach, highlighting the importance of integrating active memory retrieval mechanisms into continual learning systems.

## 1 Introduction

Deep neural networks often suffer from a critical limitation known as catastrophic forgetting: performance on previously learned tasks degrades as the model learns to perform newer ones (McCloskey & Cohen, 1989). This problem arises from the stability-plasticity dilemma (Abraham & Robins, 2005) where a model must be flexible enough to learn newer tasks while simultaneously being stable enough to retain existing knowledge. This seriously affects the ability of a model to perform well in real-world scenarios where the model must adapt to newer data and tasks incrementally over a period of time.

This paper aims to test an approach inspired by effective, synergistic learning strategies employed by humans, such as Active Recall (Roediger & Karpicke, 2006), Deliberate Practice (Ericsson et al., 1993), and Spaced Repetition (Cepeda et al., 2008). Active Recall involves effortfully retrieving information from memory. Deliberate Practice involves studying a task until a certain level of proficiency is achieved. Spaced Repetition refers to reviewing information at increasing intervals with the interval length determined by how well the task has been mastered. For example, if a student is learning Japanese, they may use flashcard applications like Anki to frequently test harder vocabulary (Active Recall), while words that are easier and recalled frequently are studied at progressively longer intervals (Spaced Repetition).

This paper proposes and evaluates a novel methodology to replicate these human learning methods within the continual learning framework. The primary contribution is Task-Focused Consolidation with Spaced Recall (TFC-SR), where the model learns new tasks while concurrently "practicing" previously learned tasks using a form of experience replay. The core of our method is the Active

Recall Probe, a periodic "memory check" where the model performs a forward pass on past experiences to evaluate its own memory state. The outcome of this memory check is then used by an adaptive spaced schedule that determines the intensity and frequency of future checks. For this initial investigation, a "naive" notion of mastery is used, where mastery refers to retained performance on previously learned tasks exceeding a predefined threshold. For the purpose of validation, TFC-SR is tested on the Split MNIST and then on the more challenging Split CIFAR-100 benchmarks and compared against standard continual learning baselines, including premier regularization-based methods like EWC (Kirkpatrick et al., 2017) and SI (Zenke et al., 2017) to demonstrate its effectiveness in alleviating catastrophic forgetting.

## 2 METHODS

### 2.1 TASK PROTOCOLS

To demonstrate the effectiveness of TFC-SR, a simple CNN and a ResNet-18 (He et al., 2016) were evaluated on two benchmark datasets, Split MNIST (Lecun et al., 1998) and Split CIFAR-100 (Krizhevsky, 2009), respectively. In both cases, after the model had learned a task, it had to classify all the classes it had seen so far. This was done to test how well the model retained its old knowledge after new information was made available to it.

Training a CNN on MNIST is a relatively simple task and is often described as the "Hello, World!" of neural networks. A basic implementation can be easily trained to achieve high accuracy on the dataset, making it suitable for testing the feasibility of TFC-SR in a relatively simple setting. MNIST consists of a total of 70,000 grayscale, $28 \times 28$ images of handwritten digits 0–9. These 70,000 images are then split into two sets: 60,000 for training and 10,000 for testing. Split MNIST is a variant of this dataset that is further divided into multiple classification tasks. In our case, it was divided into five tasks, each involving the model learning and classifying a pair of digits. This setup was used to demonstrate continual learning, in which the model receives data sequentially rather than having access to the entire training set at once.

CIFAR-100 was chosen to represent a better, relatively more difficult benchmark after TFC-SR demonstrated its effectiveness within a simple setting. CIFAR-100 consists of 60,000 color, $32 \times 32$ images representing 100 classes. There are 600 images per class and the 100 classes are grouped into 20 different superclasses. Similar to MNIST, CIFAR-100 is also split into training set and testing set with each containing 50,000 and 10,000 images respectively. Since this was going to be too difficult for a simple CNN to handle, a ResNet-18 was chosen as the model of choice. For our purposes, CIFAR-100 was divided into 10 tasks, with 10 classes per task.

### 2.2 MODEL ARCHITECTURE

All models were implemented using PyTorch and trained from scratch. For the first benchmark of Split MNIST, a standard CNN was used with convolutional blocks followed by a classifier head. The first convolutional block consisted of a 2D convolutional layer with 32 $3 \times 3$ kernels followed by a ReLU activation function and a $2 \times 2$ max-pooling layer. The second convolutional block was identical to the first one, with the only difference being that the 2D convolutional layer had 64 $3 \times 3$ kernels. The output of these blocks was flattened and passed to the classifier head, which comprised two fully connected layers. The first fully connected layer had 128 units with ReLU activation while the second one was a linear layer with 10 units corresponding to the 10 digit classes.

For the Split CIFAR-100 benchmark, a ResNet-18 model was used. To adapt the model to CIFAR-100's $32 \times 32$ image size, two conventional modifications were made: the initial $7 \times 7$ convolutional layer was replaced with a $3 \times 3$ convolutional layer, and the subsequent max-pooling layer was removed by replacing it with an identity mapping. The final fully connected layer was replaced with a linear layer containing 100 units, corresponding to the 100 classes in the CIFAR-100 dataset.

### 2.3 LEARNING ALGORITHMS AND BASELINES

For both Split MNIST and Split CIFAR-100, TFC-SR was compared against the following methods:

- Sequential Fine-tuning (Baseline): This baseline represents a lower-bound reference without any additional method and was used to evaluate the benefit of applying TFC-SR.

- Standard Experience Replay (ER) (Mnih et al., 2015): In this method, the model has access to data from previous tasks. Since TFC-SR is also a replay-based approach, Standard ER serves as a reference for assessing the contribution of the Active Recall Probe.

- Elastic Weight Consolidation (EWC) (Kirkpatrick et al., 2017): A leading regularization-based method that adds a penalty term to protect weights that are important for previous tasks, as determined by the Fisher Information Matrix.

- Synaptic Intelligence (SI) (Zenke et al., 2017): This is another prominent regularization-based method. SI protects weights that have contributed significantly to reducing the loss during training on prior tasks. The regularization strength, $\lambda$, was determined via coarse hyperparameter tuning for both EWC and SI.

## 2.4 TFC-SR ALGORITHM

TFC-SR extends the standard experience replay framework using an adaptive scheduling mechanism based on Active Recall and Spaced Repetition. The training process for each task consists of two main components: continuous mixed-batch training and an adaptive active recall schedule.

Continuous Mixed-Batch Training: for all tasks, the model is trained on mixed batches of data. Each batch is composed of a 50/50 split of new examples from the current task and examples randomly sampled from a replay buffer containing data from all the past tasks. To maintain a diverse and representative memory of the past experiences within a fixed memory budget, a reservoir replay buffer of fixed capacity was used.

Active Recall Probe and Adaptive Scheduling: The core novelty of TFC-SR is its adaptive scheduling mechanism, which we term the Active Recall Probe. This probe is a "memory check" performed periodically after each training epoch. It involves evaluating the model's performance on a batch of samples drawn from the replay buffer. To obtain an accurate measure of retained knowledge in a class-incremental setting, this evaluation is conducted in a task-aware manner, where the model's output logits are masked to include only the classes currently present in the replay buffer. We refer to this evaluation metric as the "average accuracy" throughout the paper, indicating the model's overall accuracy on the combined set of all seen tasks, rather than an arithmetic mean of accuracies computed separately per task.

The outcome of the probe dictates the schedule for the next one. If the model's performance on the replay buffer exceeds a predefined mastery_threshold, its memory is deemed stable and the time until the next probe (replay_gap) is increased by a multiplicative factor. Otherwise, if the performance is below the threshold, the model's memory is considered weak and the next probe is scheduled for the very next epoch to encourage more intensive consolidation. The full procedure for training on a single task (i.e., one experience) is detailed below in Algorithm 1.

### 2.4.1 ALGORITHM 1

```
Require: Model m, Optimizer o, Criterion c, Task Experience e
Require: Replay Buffer b
Require: Hyperparameters:
    epochs, mastery_threshold, initial_gap, gap_multiplier

add new data from e to b
l = DataLoader(e.dataset)
replay_gap = initial_gap
replay_timer = replay_gap

for epoch from 1 to epochs do
    m.train()
        for new_data, new_targets in l do
```

```
            // here, sufficient means there is enough data for
            // a 50/50 mixed batch
            if len(b) is sufficient then
                old_data, old_targets = b.sample()
                mixed_data = concat(new_data, old_data)
                mixed_targets = concat(new_targets, old_targets)

                // TrainStep: performs one gradient update using
                // the given inputs and targets
                TrainStep(m, o, c, mixed_data, mixed_targets)
            end if
        end for

    if epoch == replay_timer and b not empty then
        // perform an active recall probe
        m.eval()
        replay_perf = evaluate_replay_buffer(m, b)

        if replay_perf >= mastery_threshold then
            replay_gap = replay_gap * gap_multiplier
            replay_timer = replay_timer + replay_gap
        else
            replay_timer = replay_timer + 1
        end if
    end if
end for
```

## 3 EXPERIMENTS AND RESULTS

### 3.1 IMPLEMENTATION DETAILS

All experiments were implemented using PyTorch and trained on a single NVIDIA V100 GPU via Google Colab. To reduce randomness and ensure fair comparisons, a fixed random seed (42) was used.

A simple CNN was used for Split MNIST, while for the more challenging Split CIFAR-100, a modified ResNet-18 architecture was used, as described in Section 2.2. Unless otherwise specified, all models were trained using the Adam optimizer with a learning rate of 0.001, determined via a hyperparameter tuning process on the baseline model. The batch size was set to 64 for both benchmarks. For Split MNIST, 10 epochs were used per task, while 20 were used for Split CIFAR-100.

The key hyperparameters for each continual learning method were selected using coarse hyperparameter tuning to ensure a competitive comparison. For our proposed TFC-SR, we analyzed its sensitivity to both the mastery threshold and the buffer capacity (see Appendix A and B for full details). As an extension, buffer size sensitivity was also measured for Standard ER. For the regularization-based methods EWC and SI, the regularization strength, $\lambda$, was also selected via coarse hyperparameter tuning over several orders of magnitude. The final values used for each method are listed in Table 1.

### 3.2 EVALUATION ON SPLIT MNIST

To validate the feasibility of TFC-SR, experiments were conducted on the Split MNIST benchmark. Figure 1 illustrates the learning curves, plotting the average accuracy on all seen tasks after each of the five sequential tasks was completed.

The results clearly demonstrate the severity of catastrophic forgetting. After training on five tasks, the performance of the baseline (Sequential Fine-tuning) model dropped to a final accuracy of

Table 1: Final hyperparameter values for main experiments.

| Benchmark | Method | Hyperparameter | Value |
|---|---|---|---|
| Split MNIST | EWC | Regularization $\lambda$ | 10000.0 |
| | SI | Regularization $\lambda$ | 100.0 |
| | Replay Based (ER, TFC-SR) | Buffer Capacity | 200 |
| | TFC-SR | Mastery Threshold | 95.0 |
| Split CIFAR-100 | EWC | Regularization $\lambda$ | 10000.0 |
| | SI | Regularization $\lambda$ | 1.0 |
| | Replay Based (ER, TFC-SR) | Buffer Capacity | 1000 |
| | TFC-SR | Mastery Threshold | 10.0 |

All replay-based methods used a replay-to-new-data ratio of 0.5 in each batch. The TFC-SR scheduler used an initial replay gap of 1 and a gap multiplier of 1.5 for all experiments.

10.28%. The regularization-based methods provided modest improvements over the baseline, with EWC achieving 20.37% accuracy and SI reaching a final accuracy of 32.58%.

A significant performance improvement was observed with the replay-based methods. The standard ER baseline achieved a final accuracy of 90.44%, showing that even non-adaptive replay is highly effective for this benchmark. The proposed TFC-SR method also achieved a strong and comparable final accuracy of 86.22%. Notably, the learning trajectory of TFC-SR was competitive with Standard ER throughout the experiment, briefly outperforming it until the third task before finishing at a similar level by the fourth task. The strong performance of TFC-SR motivated further analysis on a more challenging dataset.

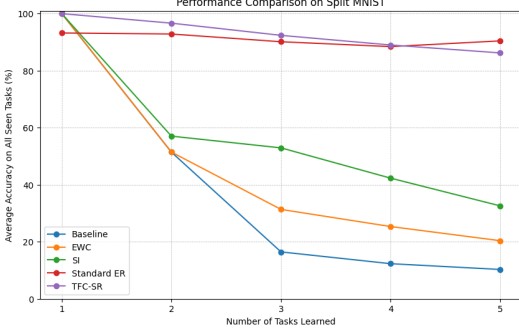

Figure 1: Performance comparison of TFC-SR against baseline methods on the Split MNIST benchmark. The plot shows average accuracy on all previously seen tasks (y-axis) after each new task is learned (x-axis). Higher is better.

### 3.3 SCALABILITY ANALYSIS ON SPLIT CIFAR-100

After the feasibility of the proposed TFC-SR was established, it was tested on a much harder Split CIFAR-100 benchmark in order to perform Scalability Analysis. The results are outlined in Figure 2. As expected, all methods experienced a significant drop in performance on this harder benchmark. Sequential Fine-tuning only managed a final performance of 7.27%. The regularization-based methods, EWC and SI, failed to provide any substantial improvement and had final average accuracy of 6.16% and 8.15% respectively. Standard ER, which showed strongest results on previous MNIST benchmark, ended up with a final average accuracy of 7.4%.

In contrast, TFC-SR maintained a distinct performance advantage throughout the 10-task sequence, achieving a final performance of 13.17%. This result, which is more than 1.6 times the

Table 2: Final average accuracy (%) after 10 tasks on Split CIFAR-100 for standard ER and TFC-SR with varying replay buffer capacities.

| Buffer Capacity | Standard ER (%) | TFC-SR(%) |
|---|---|---|
| 100 | 8.51 | 8.55 |
| 500 | 9.31 | 9.48 |
| 1000 | 7.40 | 13.17 |
| 2000 | 21.37 | 19.72 |

next best method (SI), demonstrates that TFC-SR's active recall mechanism scales effectively to more complex and challenging continual learning scenarios.

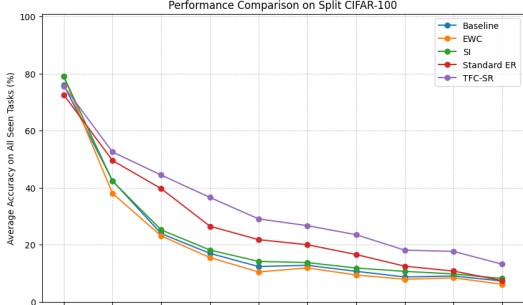

Figure 2: Performance comparison of TFC-SR against baseline methods on the Split CIFAR-100 benchmark. The plot shows the average accuracy on all tasks seen so far (y-axis) after each new task is learned (x-axis). Higher is better.

### 3.4 ABLATION STUDY: EFFECT OF BUFFER CAPACITY

During the initial hyperparameter tuning for TFC-SR on the Split CIFAR-100 benchmark, it was observed that the model's performance on memory checks was consistently high, suggesting that the memory retention task was not sufficiently challenging when using a relatively large buffer. To investigate the behavior of our method under varying degrees of memory pressure, an ablation study was conducted on the replay buffer capacity. The performance of TFC-SR against Standard ER was compared across four buffer sizes: 100, 500, 1000, and 2000. The complete learning curves for each run are detailed in Appendix B, while the final accuracies are summarized in Table 2 and visualized as a trend plot in Figure 3, with the buffer capacity shown on a logarithmic scale. The results reveal a clear and meaningful trade-off between the two methods.

In low-memory (capacity = 100) and medium-memory (capacity = 500, 1000) settings, TFC-SR consistently outperforms Standard ER. The performance advantage is greater at a capacity of 1000, where TFC-SR achieves a final accuracy of 13.17% against Standard ER's 7.40%. This suggests that when memory resources are constrained, the Active Recall Probe and adaptive scheduling of TFC-SR provide a significant benefit over a non-adaptive replay strategy.

However, in the high-memory setting (capacity = 2000), this trend reverses. Standard ER achieves a final accuracy of 21.37%, surpassing TFC-SR's 19.72%. This outcome suggests that in high-memory settings, the diversity of the buffer alone is sufficient to maintain performance, reducing the relative advantage of TFC-SR's Active Recall Probe and adaptive scheduling mechanisms.

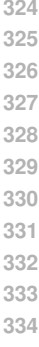

Figure 3: Effect of replay buffer capacity on final average accuracy. The plot shows the final average accuracy (y-axis) after training on 10 tasks for Standard ER and TFC-SR across varying buffer capacities (x-axis, log-scaled base 10) on the Split CIFAR-100 benchmark. Higher values indicate better performance.

Table 3: Efficiency comparison of replay-based methods on Split CIFAR-100 benchmark. All methods use a buffer capacity of 1000.

| Method | Total Replay Batches | Memory Checks | Final Accuracy (%) |
|---|---|---|---|
| Standard ER | 15800 | 0 | 7.40 |
| TFC-SR | 15800 | 50 | 13.17 |
| TFC-SR (Spaced Replay) | 3555 | 45 | 10.18 |

## 3.5 EFFICIENCY ANALYSIS

In addition to accuracy, the computational efficiency of the replay-based methods on the Split CIFAR-100 benchmark was recorded using the total number of replay batches performed during training. For TFC-SR, the total number of memory checks performed was also measured. The results are summarized in Table 3 for the primary experiments (buffer_capacity = 1000) as well as for the "Spaced Replay" variant of TFC-SR.

In the main experiments, both TFC-SR and Standard ER performed an identical number of replay batches (15,800) with the only difference being the 50 memory checks performed by TFC-SR. This indicates that the performance advantage of TFC-SR (13.17% vs. 7.40%) stems from the stabilizing effect of its Active Recall Probe and adaptive scheduling, rather than the amount of replay volume alone.

The trade-off between replay volume and accuracy was further examined in Appendix C, where a variant of TFC-SR with Spaced Replay achieved competitive performance of 10.18%, surpassing Standard ER, while reducing the number of replay batches by over 77%. This highlights the potential for significant computational savings when an adaptive schedule is allowed to reduce the replay frequency.

## 4 DISCUSSION

As detailed in Section 3, TFC-SR was compared against other prominent continual learning methods on two benchmarks: Split MNIST and Split CIFAR-100. On both benchmarks, it demonstrated strong performance and significantly outperformed all other methods on the more challenging Split CIFAR-100 benchmark. These results indicate that TFC-SR is a highly effective strategy for continual learning. The results from the ablation studies further revealed a key trade-off, showing that TFC-SR's advantage is more pronounced in memory-constrained settings.

The performance gain of TFC-SR over Standard ER was achieved despite both methods performing an identical number of replay batches and the only difference being the presence of the active recall mechanism in TFC-SR. This suggests that the benefit stems not from the volume of replay, but from the act of memory checking itself. We refer to this mechanism as the "Active Recall Probe". We posit two primary reasons for its stabilizing effect, whose contributions appear to depend on the model architecture. Our primary hypothesis is that repeated forward passes on past experiences compel the network's Batch Normalization layers to maintain their running statistics for older distributions, directly countering a known source of representational drift. This is strongly supported by our results on the Split CIFAR-100 benchmark, where the ResNet-18 model (which contains Batch Normalization) showed a dramatic performance advantage for TFC-SR over Standard ER. Our secondary hypothesis is that periodic activation of neural pathways associated with past memories acts as a form of implicit regularization, strengthening those representations against interference. The results from the Split MNIST benchmark, which used a simple CNN without Batch Normalization layers, are consistent with this view. On this benchmark, TFC-SR's advantage was minimal, suggesting that the implicit regularization effect alone provides a more subtle, though still present, benefit. Therefore, we believe that it is the synergy between this implicit regularization and the stabilization of Batch Normalization that explains TFC-SR's significant advantage over the Standard Experience Replay baseline on the Split CIFAR-100 benchmark. Additionally, this mechanism is extremely efficient and requires minimal computational resources. As mentioned in section 3.5, it only required 50 probes for the entire Split CIFAR-100 benchmark.

TFC-SR contributes a novel mechanism to the family of replay-based continual learning methods. Unlike premier regularization methods such as EWC (Kirkpatrick et al., 2017) and SI (Zenke et al., 2017), which indirectly protect old knowledge by penalizing weight changes, TFC-SR directly reinforces past representations through its Active Recall Probe. Our approach also differs from other advanced replay methods. While excellent methods like iCaRL (Rebuffi et al., 2017) focus on intelligent exemplar management strategies for replay buffer that prioritize informativeness and notable generative approaches like those in (Van De Ven et al., 2020) and (Tadros et al., 2022) focus on creating replay data without storage, TFC-SR introduces a new dimension: the process of active, periodic memory retrieval as a learning mechanism in its own right.

Nonetheless, this study is not without its own limitations. First, our experiments were conducted exclusively on image classification benchmarks; the efficacy of TFC-SR on other data modalities such as natural language processing or reinforcement learning tasks remains to be explored. Second, our replay buffer used a standard reservoir sampling strategy. Incorporating more advanced buffer management techniques, such as iCaRL's herding algorithm, Iterative Projection and Matching (IPM) (Zaeemzadeh et al., 2019) or gradient-based sample selection (Aljundi et al., 2019) could provide further improvements. Third, our adaptive scheduler relied on a naive notion of "mastery". It was based on a simple performance threshold. A more nuanced metric could unlock more sophisticated and efficient scheduling dynamics. Lastly, this work was conducted independently without access to large-scale infrastructure. As a result, we focused on feasible settings with modest compute requirements. Future work could explore scaling TFC-SR to more diverse architectures and longer task sequences.

The findings in this paper open several exciting avenues for future research. Our preliminary results with a "Spaced Replay" variant (Appendix C) show a promising trade-off between accuracy and computational cost, warranting a deeper investigation into sparse, episodic consolidation. Furthermore, our experiments with a performance-threshold-based progression baseline (Appendix D) — a simple, literal interpretation of deliberate practice — underscore the potential of adaptive task scheduling and self-directed learning strategies as fruitful directions for future work. Ultimately, our results support the hypothesis that building computational analogues of human cognitive processes, such as Active Recall, is a fruitful direction for the development of more robust, efficient, and capable artificial intelligence systems.

## REPRODUCIBILITY STATEMENT

We have made every effort to ensure the reproducibility of our results. The complete source code for all experiments is provided in the supplementary material as a zip file. This includes Jupyter notebooks to both run the experiments from scratch (`experiments_mnist.ipynb` and

experiments_cifar100.ipynb) and generate all figures and tables from our precomputed results (plots_and_results.ipynb). All software dependencies and their exact versions are listed in the requirements.txt file. Please note that this file was generated in the Google Colab environment where the experiments were carried out. The precise details of our model architectures are described in Section 2.2. The final tuned hyperparameters for all methods, including the random seed (42), are provided in the "Implementation Details" subsection (Section 3.1) and its corresponding Table 1. All experiments were conducted on standard, publicly available benchmarks (Split MNIST and Split CIFAR-100), and we used the popular Avalanche library to create the continual learning task splits, as detailed in Section 2.1, ensuring the data protocol is easily reproducible.

## USE OF LARGE LANGUAGE MODELS

We acknowledge the significant contribution of a Large Language Model (LLM) in the preparation of this manuscript, in accordance with the ICLR policy. Specifically, the LLM was used to refine readability, correct grammatical structure, and polish writing. It also served as a tool for research ideation and brainstorming in the early stages of this project, and an LLM-based code assistant helped save significant time during the experimental phase.

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

## A  TFC-SR'S SENSITIVITY TO MASTERY THRESHOLD

To find the optimal mastery threshold for TFC-SR, we tested its performance across a range of values on the Split CIFAR-100 benchmark: (10.0, 20.0, 30.0, 50.0, 70.0, 90.0, 99.0). We found that the performance was robust across all values. The results are summarized in Table A.1 and Figure A.1.

For thresholds between 10% and 90%, the final accuracy was nearly identical (13.17%), indicating that the model's performance on the replay buffer consistently exceeded the targets. This led the adaptive scheduler to adopt its most efficient mode by performing the minimum number of memory checks (50–63). The 70% threshold run yielded a slightly lower accuracy of 12.45% with 51 memory checks, likely due to stochastic variation from random sampling during training.

The 99% threshold served as a crucial stress test, creating a condition where the memory check would only succeed if the model's retention of past knowledge was nearly perfect. The scheduler performed 103 memory checks in this configuration, significantly more than in the lower-threshold runs. This confirms that the check was frequently failing, thereby triggering the intended increase in probing.

However, the fact that only 103 checks were performed, roughly half of the maximum possible, proves that the model did succeed in meeting the 99% threshold on multiple occasions. This indicates instances of near-perfect memory consolidation even while learning new tasks.

Table A.1: TFC-SR's performance across multiple mastery thresholds on CIFAR-100 benchmark.

| Mastery Threshold | Memory Checks | Final Accuracy (%) |
|---|---|---|
| 10.0 | 50 | 13.17 |
| 20.0 | 50 | 13.17 |
| 30.0 | 50 | 13.17 |
| 50.0 | 50 | 13.17 |
| 70.0 | 51 | 12.45 |
| 90.0 | 63 | 13.17 |
| 99.0 | 103 | 11.58 |

In particular, the final accuracy of this 99% threshold run was 11.58%. This result significantly outperforms the Standard ER baseline (7.40%), highlighting the benefit of the Active Recall Probe mechanism. Additionally, this performance is only slightly lower than that of the best-performing threshold values (13.17%), showing that the algorithm is robust and does not catastrophically fail even when subjected to a highly demanding and volatile replay schedule.

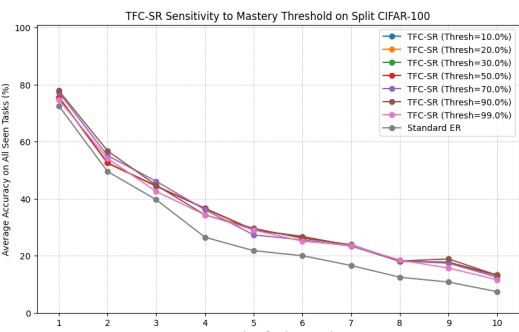

Figure A.1: Learning curves of TFC-SR with varying mastery thresholds on Split CIFAR-100. For comparison, the performance of Standard ER baseline (buffer capacity = 1000) is included. The plot shows the average accuracy on all tasks seen so far (y-axis) after each new task is learned (x-axis).

## B    FULL LEARNING CURVES FOR BUFFER CAPACITY ANALYSIS

This section provides the detailed learning curves that support the ablation study on replay buffer capacity presented in Section 3.4.

We evaluated both TFC-SR and the Standard ER baseline across four different buffer capacities: 100, 500, 1000, and 2000. The complete task-by-task learning trajectories for each TFC-SR run are shown in Figure B.1. The corresponding trajectories for the Standard ER baseline are shown in Figure B.2. The final accuracy values from these curves were used to generate the trend plot in Figure 3 and the summary in Table 2 of the main paper.

## C    ANALYSIS OF A SPACED REPLAY VARIANT (TFC-SR²)

To further investigate the trade-off between replay volume and final performance, we implemented and tested a "Spaced Replay" variant of TFC-SR, which we informally named TFC-SR$^2$. In this setup, continuous mixed-batch replay is disabled; instead, the replay is performed only during the specific epochs that are triggered by the adaptive scheduling mechanism. In all other epochs, the model trains exclusively on data from the current task. This experiment allows us to isolate the impact of the scheduled replay events themselves under a drastically reduced computational budget.

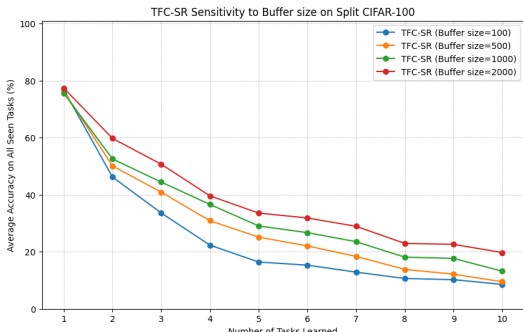

Figure B.1: Learning curves of TFC-SR with varying buffer capacities on Split CIFAR-100. The plot shows the average accuracy on all tasks seen so far (y-axis) after each new task is learned (x-axis).

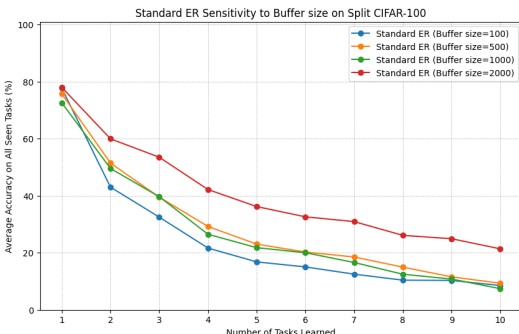

Figure B.2: Learning curves of Standard ER with varying buffer capacities on Split CIFAR-100. The plot shows the average accuracy on all tasks seen so far (y-axis) after each new task is learned (x-axis).

The results of this experiment are compared with our primary TFC-SR baseline and the Standard ER baseline in Table 3 of the main paper, while the full learning curves are illustrated in Figure C.1.

The TFC-SR (Spaced Replay) variant achieved a final accuracy of 10.18%. This result is significant for two reasons. First, despite using over 77% fewer replay batches (3,555 vs. 15,800), it substantially outperforms the brute-force Standard ER baseline (7.40%), demonstrating that even a small amount of intelligently scheduled replay is more effective than high-volume, non-adaptive replay. Second, while its accuracy is lower than our main continuous replay variant (13.17%), this result confirms that a higher replay volume remains beneficial for achieving maximum performance on a challenging benchmark. This finding highlights a clear and practical trade-off between computational efficiency and performance.

## D    MASTERY-GATED PROGRESSION

To explore a more literal interpretation of the deliberate practice analogy, we implemented a performance-gated, curriculum-style baseline, which we refer to as Mastery-Gated Progression (MGP) for convenience. While conceptually similar to existing curriculum and self-paced learning strategies, MGP is adapted to our continual learning setting. In contrast to TFC-SR's interleaved training and replay, MGP separates them into distinct phases: the model trains on each task until it surpasses a predefined mastery threshold, at which point it progresses to the next task. This results in an adaptive schedule where training time on each task emerges from its inherent difficulty.

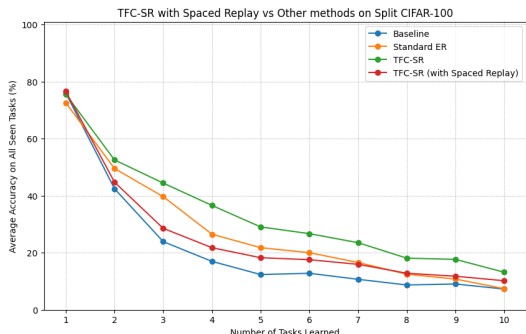

Figure C.1: Learning curves comparing TFC-SR (Spaced Replay) against our primary TFC-SR model and the Standard ER baseline on Split CIFAR-100. All methods used a buffer size of 1000. The plot shows the average accuracy on all tasks seen so far (y-axis) after each new task is learned (x-axis).

MGP uses a two-part mastery criterion: the model must achieve a test accuracy above a mastery_threshold on the current task, while maintaining performance above a retention_threshold on previously seen tasks. To prevent indefinite training on particularly difficult tasks, a max_epochs parameter is used as a safeguard. The full procedure is outlined in Algorithm D.1.

We tested MGP on the Split CIFAR-100 benchmark. Its performance is compared with the primary baselines in Figure D.1. The MGP model achieved a final average accuracy of 11.63%, outperforming the Standard ER baseline (7.40%). While not surpassing our main method (TFC-SR), this result suggests that performance-gated progression is a viable training strategy and merits further exploration.

The most insightful finding from this experiment was the variable number of epochs required to achieve mastery for each task: (47, 9, 22, 50, 50, 9, 15, 18, 7, 19). This result makes explicit the inherent difficulty of each incremental task. For example, the model required 47 epochs to learn the first task from a random initialization, but only 9 for the second, suggesting a degree of positive knowledge transfer. This dynamic allocation of training effort, driven by task difficulty and retention, contrasts with fixed-epoch schedules and highlights the potential for more adaptive, self-directed approaches (Zhu et al., 2022) to continual learning.

ALGORITHM D.1   MASTERY GATED PROGRESSION (MGP) TRAINING FOR A SINGLE TASK

```
Require: Model m, Optimizer o, Criterion c, Task Experience e
Require: Replay Buffer b, task_id
Require: Hyperparameters:
    max_epochs, mastery_threshold, retention_threshold

epoch = 0
mastery_achieved = false
l = DataLoader(e.dataset)

while not mastery_achieved and epoch < max_epochs do
    epoch = epoch + 1
    m.train()

    for new_data, new_targets in l do
        if task_id > 0 then
            // perform mixed-batch training
            old_data, old_targets = b.sample()
            mixed_data = concat(new_data, old_data)
            mixed_targets = concat(new_targets, old_targets)
```

```
                    TrainStep(m, o, c, mixed_data, mixed_targets)
            else
                // perform new-task-only training
                TrainStep(m, o, c, new_data, new_targets)
            end if
        end for

        // perform mastery check
        m.eval()
        new_task_perf = evaluate(m, e.test_dataset)
        retention_perf = evaluate_replay_buffer(m, b)

        if new_task_perf >= mastery_threshold and
            retention_perf >= retention_threshold then
            mastery_achieved = True
        end if
    end while
    return epoch
```

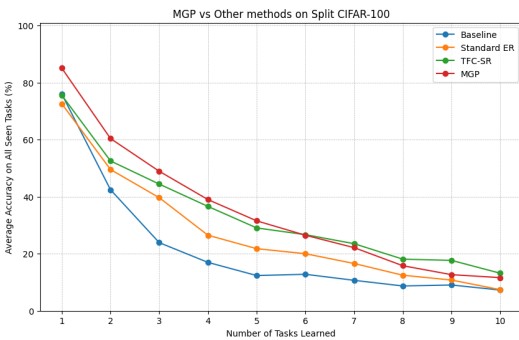

Figure D.1: Learning curves comparing MGP with TFC-SR and the Standard ER on Split CIFAR-100. All methods used a buffer size of 1000. The plot shows the average accuracy on all tasks seen so far (y-axis) after each new task is learned (x-axis).

