# OpenReview forum: "Task-Focused Consolidation with Spaced Recall: Making Neural Networks Learn like College Students"
_ICLR.cc/2026/Conference — ICLR 2026 Conference Withdrawn Submission_

### Official Review · Reviewer_s9yp · 2025-10-27

**Soundness:** 1
**Presentation:** 2
**Contribution:** 1
**Rating:** 2
**Confidence:** 5

**Summary:**

Continual Learning involves training a model on a sequence of tasks, with the main goal of learning new tasks without forgetting previously learned ones. This paper presents an approach inspired by 3 human learning strategies. (1) Active Recall (which retrieves previous information), (2) Deliberate Practice (learn until proficiency is achieved), and (3) Spaced Repetition (add space between retrieving information). The paper evaluates the proposed method on 2 standard benchmarks (Split MNIST and CIFAR100) and compares it with classical methods (EWC, SI, and ER). The paper also presents ablation experiments on the size of the memory buffer.

**Strengths:**

- The authors motivate the idea of incorporating brain-inspired strategies into the training of Deep Learning methods. One of the challenges of current Deep Learning models is their limited ability to accumulate knowledge, something that happens naturally in humans. Human learning seems a natural source of inspiration for proposing new approaches to tackle this challenge.
- Section 4 (Discussion) provides a clear explanation of the results and their comparison with other methods. Showing the conclusions together with explanations for their decisions.

**Weaknesses:**

- There is an evident lack of understanding of previous methods in the area of Continual Learning. For example, the paper's "Mixed-Batch" strategy, which it calls novel, is a common practice in the area.
    - Chrysakis, Aristotelis, and Marie-Francine Moens. "Online continual learning from imbalanced data." International Conference on Machine Learning. PMLR, 2020.
- It is unclear how and when the Adaptive Active Recall is used during training. Algorithm 1 shows only how it is used during inference, so it only affects how accuracy is calculated.
    - There is no ablation on how this can affect the final results of the experiments.
- The experiments section lacks more complex scenarios. MNIST and CIFAR100 are standard in the area but are mostly outdated due to their simplicity. More complex scenarios are needed, such as Tiny ImageNet, CORE50, or others.
- The only replay-based method is Standard Replay, which has some weird results in Figure 3. Why does it decrease when using 1000 samples? Shouldn't it show an increase in performance?
- More and better baselines are needed. Multiple methods use memory to tackle the challenge of Catastrophic Forgetting. As mentioned in the paper, iCarl is one of them, but others, such as DER, can also be used for comparison.
    - Buzzega, Pietro, et al. "Dark experience for general continual learning: a strong, simple baseline." Advances in neural information processing systems 33 (2020): 15920-15930.
    - Depending on how the authors describe the method, comparing it with other memory-based methods is not adequate, because the proposed method can complement any such method. Similar to this idea are storage policy methods, for example:
        - Hurtado, Julio, et al. "Memory population in continual learning via outlier elimination." Proceedings of the IEEE/CVF International Conference on Computer Vision. 2023.

**Questions:**

- Did you use previous implementations or frameworks to run experiments of the previous methods, or did you implement them from scratch?
    - This question concerns results from previous methods, which are lower than those reported in other papers.

---

### Official Review · Reviewer_3dJN · 2025-10-28

**Soundness:** 1
**Presentation:** 2
**Contribution:** 1
**Rating:** 0
**Confidence:** 5

**Summary:**

The paper proposes a Continual Learning approach inspired by how a college student learns. In a nutshell, the method emphasizes the repetition of past concepts and the careful selection of intervals between these repetitions. In practice, it builds upon Experience Replay (a well-established baseline in the field) by introducing a scheduling mechanism that determines when examples from the memory buffer are replayed. The authors evaluate their method on Split MNIST and Split CIFAR-100, reporting mixed results.

**Strengths:**

Standard rehearsal approaches typically interleave batches of examples from the current task with batches retrieved from the memory buffer. In contrast, this work proposes replacing this regular and constant scheduling with an adaptive mechanism, where rehearsal is triggered only when accuracy on past tasks decreases. This is a sound and promising intuition: such an adaptive strategy could reduce unnecessary computation (e.g., memory retrieval and additional forward–backward passes) while mitigating the risk of overfitting to the limited set of samples stored in the buffer, a well-known issue in rehearsal-based methods.

**Weaknesses:**

The paper has countless problems in its current form. The impression is that the paper is an exercise for a student to practice submitting to top conferences. In general, the writing is good, but it seems that an LLM did much of the work, as also corroborated by the statement placed by the authors on the last page. These are some of the major points I would like to point out:
- The motivation is unclear. The authors make a proposal, which has some intuitive explanation inspired by how a student learns, but it is not clear how this proposal can improve upon well-established approaches.
- The experimental section is really poor. The authors perform experiments on Split MNIST and CIFAR-100, with not very good results. Many competitors are missing (e.g., DER++, iCaRL, and so on), and the results on CIFAR-100 (around 10–15% accuracy) are not consistent with previous papers or with common sense.
- The experimental comparison with Experience Replay should be clearer and more thorough. Since the major contribution of this paper is the scheduling approach, it must be shown clearly that this modification leads to substantial improvements across multiple settings. Actually, this is hard to conclude when looking at the tables.

**Questions:**

No questions.

---

### Official Review · Reviewer_Nqjw · 2025-11-01

**Soundness:** 3
**Presentation:** 2
**Contribution:** 2
**Rating:** 4
**Confidence:** 5

**Summary:**

This paper introduces a continual learning method, Task-Focused Consolidation with Spaced Recall (TFC-SR), inspired by human learning strategies such as Active Recall and Spaced Repetition. The proposed method extends the standard experience replay (ER) framework with a novel mechanism called the "Active Recall Probe." This probe consists of a periodic, no-gradient forward pass on a batch of replay samples to evaluate the model's memory of past tasks. Based on whether the performance on this probe exceeds a predefined "mastery threshold," an adaptive schedule increases or decreases the interval until the next probe. The method is evaluated on Split MNIST and Split CIFAR-100. On Split MNIST, TFC-SR's performance is comparable to standard ER. On Split CIFAR-100 with a buffer of 1000 samples, TFC-SR significantly outperforms ER (13.17% vs. 7.40%). However, an ablation study reveals that this advantage is specific to memory-constrained settings and disappears when a larger buffer is used, where standard ER becomes superior.

**Strengths:**

1. Intuitive and Simple Concept: The motivation drawn from human cognitive science—specifically Active Recall and Spaced Repetition—provides a compelling and intuitive narrative for the method.
2. Strong Performance in a Specific Regime: The paper demonstrates a significant performance advantage for TFC-SR in a memory-constrained setting on a challenging benchmark.
3. Insightful Ablation and Discussion: The ablation study on buffer capacity (Section 3.4) is a highlight of the paper.

**Weaknesses:**

1. Major Disconnect Between Proposed Mechanism and Likely Cause: The paper's central narrative is built around the cognitive science concepts of "Active Recall" and "Spaced Repetition." However, the authors' own analysis in Section 4 strongly suggests that the observed performance gain on Split CIFAR-100 is an architectural artifact related to Batch Normalization.
2. Inconsistent and Underwhelming Empirical Results: The experimental results are mixed and do not support a general claim of the method's superiority.
3. Limited Algorithmic Novelty: The core algorithm is a very incremental1 modification of experience replay.

**Questions:**

1. Your discussion section compellingly argues that the performance gain on Split CIFAR-100 is likely due to the stabilization of Batch Normalization statistics by the probe's forward passes. To definitively test this hypothesis, have you considered replacing the BN layers in the ResNet-18 model with an alternative normalization scheme that does not use running statistics, such as Group Normalization or Layer Normalization?
2. The results demonstrate that TFC-SR's advantage is highly sensitive to the replay buffer size, with standard ER being superior on Split MNIST and also on Split CIFAR-100 with a large buffer

---

### Note · Authors · 2025-11-22

**Comment:**

We would like to thank all reviewers for their time and effort. We have decided to withdraw our submission to allow ourselves more time to incorporate the reviewer feedback. Furthermore, we have done some additional experiments and analysis since our submission to the venue and have better results and benchmarks we want to add to the paper. We also have some modifications to the proposed method like changes in buffer population strategies. For clarification and in service of the random internet travellers of future, we would like to also answer some of the reviewer queries:

1. One motivation we had in conducting our work was to focus on method that works well in low resource environments. We wanted to prioritize low resource use even if that comes with some performance tradeoff. Thus, we conducted all our experiments in a single Google Colab notebook without using any additional compute resources other than the GPU provided by Colab as well as low using low compute time. All benchmarks that were chosen with that focus in mind. The setting is also the reason why the performance of methods in our experiments is lower than their performances in other works.

2.During training, the recommendation is to allow the adaptive schedule to also schedule replay. This provides significant compute savings with marginal performance tradeoff. As outlined in the paper, this allowed us to reduce the total amount  of replay batches by 77%.

3. The work does not claim that "Mixed-Batch" training is novel, rather the novelty claim is about adaptive scheduling of said training governed by a mastery metric.

4. The empirical results were surprising even for us like the sudden performance drop of standard ER within the 1000 buffer size regime. The weirdness was one reason we decided to include them as-is in our paper. We believe weird results in weird settings deserve as much love and attention as great results in great settings.

5. We used the Avalanche library's implementation of Split MNIST, Split CIFAR-100, EWC and SI. However, the evaluation functions and all other code (as given in the supplementary materials) was written from scratch.

6. We did not include more complex scenarios because we did not feel like they were needed just yet as the performances of all the methods used in our paper in simpler settings were nowhere close to what is ideal (as close to 100% as possible).

7. To test our hypothesis (more like an intuitive guess) about the Batch Norm layer playing a part in the results, we did try replacing them with Group Norm. The results were weird as expected with Standard ER surpassing TFC-SR on one task, only for the lead to flip in the next task, and so on. We have no idea where this behaviour comes from other than the divine benevolence of Arceus.

8. During writing the paper, the LLM was used to adjust the phrasing and Grammar of the paper. The draft of the paper was written completely by the authors from scratch before using an LLM for some glow-up. This was done to make sure the paper adheres to a more neutral, standard writing style as opposed to the author's own writing style full of individualistic, hit-or-miss quirks (according to third party assessments and feedback).

**Withdrawal Confirmation:**

I have read and agree with the venue's withdrawal policy on behalf of myself and my co-authors.